# Silibinin Downregulates Types I and III Collagen Expression via Suppression of the mTOR Signaling Pathway

**DOI:** 10.3390/ijms241814386

**Published:** 2023-09-21

**Authors:** Sooyeon Choi, Seoyoon Ham, Young In Lee, Jihee Kim, Won Jai Lee, Ju Hee Lee

**Affiliations:** 1Department of Dermatology & Cutaneous Biology, Research Institute, Yonsei University College of Medicine, Seoul 03722, Republic of Korea; choisy429@yuhs.ac (S.C.); hsy7852@yuhs.ac (S.H.); ylee1124@yuhs.ac (Y.I.L.); 2Scar Laser and Plastic Surgery Center, Yonsei Cancer Hospital, Seoul 03722, Republic of Korea; mygirljihee@yuhs.ac (J.K.); pswjlee@yuhs.ac (W.J.L.); 3Department of Dermatology, Yongin Severance Hospital, Yonsei University College of Medicine, Yongin 16995, Republic of Korea; 4Department of Plastic Surgery, Yonsei University College of Medicine, Seoul 03722, Republic of Korea

**Keywords:** keloid scar, fibrosis, mammalian target of rapamycin (mTOR), natural compound, flavonoid, silibinin

## Abstract

Keloid scars are fibro-proliferative conditions characterized by abnormal fibroblast proliferation and excessive extracellular matrix deposition. The mammalian target of the rapamycin (mTOR) pathway has emerged as a potential therapeutic target in keloid disease. Silibinin, a natural flavonoid isolated from the seeds and fruits of the milk thistle, is known to inhibit the mTOR signaling pathway in human cervical and hepatoma cancer cells. However, the mechanisms underlying this inhibitory effect are not fully understood. This in vitro study investigated the effects of silibinin on collagen expression in normal human dermal and keloid-derived fibroblasts. We evaluated the effects of silibinin on the expressions of collagen types I and III and assessed its effects on the suppression of the mTOR signaling pathway. Our findings confirmed elevated mTOR phosphorylation levels in keloid scars compared to normal tissue specimens. Silibinin treatment significantly reduced collagen I and III expressions in normal human dermal and keloid-derived fibroblasts. These effects were accompanied by the suppression of the mTOR signaling pathway. Our findings suggest the potential of silibinin as a promising therapeutic agent for preventing and treating keloid scars. Further studies are warranted to explore the clinical application of silibinin in scar management.

## 1. Introduction

Keloid scars are fibroproliferative conditions characterized by the dysregulated proliferation of dermal fibroblasts and excessive deposition of extracellular matrix (ECM) components, such as types I and III collagen, fibronectins, and glycosaminoglycans [1]. Pathological scars can cause severe itching, pain, and contractures, resulting in a lower quality of life [2]. Various strategies have been attempted to treat keloid scars, including surgery, cryotherapy, radiotherapy, the intralesional injection of triamcinolone, and chemotherapeutic agents such as bleomycin and 5-fluorouracil. However, nontargeted therapies often lead to suboptimal results and high recurrence [3]. Various cytokines, such as transforming growth factor-β1 (TGF-β1), and growth factors, such as vascular endothelial growth factor, insulin-like growth factor, and platelet-derived growth factor, are thought to promote keloid formation [4]. Recently, the Mammalian target of the rapamycin (mTOR) pathway has been suggested as a key regulator of collagen expression in human dermal fibroblasts (HDFs) and as a target for managing keloid disease. mTOR is a 289 kDa serine–threonine kinase that regulates cell proliferation, metabolic processes, protein synthesis, and autophagy [5]. The mRNA and protein levels of mTOR are significantly higher in keloid-derived fibroblasts (KFs) than those in normal HDFs, and keloid tissue extracts show the increased expression and phosphorylation of downstream molecules of mTOR, suggesting activation of the mTOR signaling pathway in keloid scars [6,7]. Rapamycin, an mTOR inhibitor, downregulates collagen expression in a dose-dependent manner in HDFs and KFs, suggesting that the inhibition of mTOR could reduce ECM deposition [6,8].

Silibinin (Figure 1) is a natural flavonoid isolated from the seeds and fruits of the milk thistle (*Silybum marianum* (L.) Gaertn; Asteraceae) [9]. Since the 1970s, silymarin (a component of milk thistle extract) has been regarded as an official medicine by the World Health Organization (WHO) for its hepatoprotective and antifibrotic properties [10,11]. In addition, silibinin, a major bioactive component of silymarin, exhibits anti-angiogenic and anti-proliferative effects in various human cancer models and inhibits the mTOR signaling pathway in human cervical and hepatoma cancer cells [9,12]. Moreover, silibinin downregulates the expression of type I collagen in HDFs; however, the mechanisms underlying this inhibitory effect are not fully understood [13].

Therefore, we hypothesized that silibinin inhibits collagen formation through the mTOR signaling pathway, potentially making it a viable therapeutic agent for scars. We aimed to investigate the effects of silibinin on collagen expression in normal human dermal and keloid-derived fibroblasts, evaluate the effects of silibinin on the expressions of collagen types I and III, and assess its effects on the suppression of the mTOR signaling pathway.

## 2. Results

### 2.1. Clinical Information of Investigated Tissue Samples

After surgical excision for keloid treatment, five keloid scar tissue samples were collected from the anterior part of the trunk or ear (Figure 2). Before the operation, none of the patients underwent scar treatment, including triamcinolone injection, cryotherapy, or laser treatment. These scars enlarged beyond the initial scar margin, invading the surrounding normal tissues. Moreover, they did not undergo regression for over a year. Three of the five scars showed recurrence after surgical excision and underwent intra-lesional triamcinolone injection.

Three normal human skin specimens were also obtained from a free transverse rectus abdominis myocutaneous (TRAM) flap. Details of the keloid and normal skin tissue samples used in this study are listed in Table 1.

### 2.2. mTOR Phosphorylation Is Elevated in Active Keloid Scar Compared to the Normal Tissue Specimen

To confirm the increased mTOR signaling in keloid tissues, we compared the expression of p-mTOR between normal and keloid tissues using IHC (Figure 3A). Semi-quantitative analysis revealed that the levels of p-mTOR increased 21.3-fold in keloid tissue compared with those in normal tissue (*p* < 0.01) (Figure 3B). Our analyses showed elevated mTOR phosphorylation in keloid scars, compared with that in normal skin tissues, especially in fibroblasts (Figure 3C).

### 2.3. Effect of Silibinin on Cell Viability in HDFs and KFs

To determine silibinin’s effect on the viability of HDF and KF cells, the cells were exposed to silibinin (0–200 μM) for 24 h. As shown in Figure 4, cell viability silibinin treatment did not affect the viability of the cells. These data indicate that a concentration of silibinin under 200 μM does not exert any cytotoxic effects on HDFs and KFs.

### 2.4. Silibinin Significantly Reduced Collagen I and III mRNA Transcripts in TGF-β1-Treated HDFs and KFs 

Upregulated collagen expression is frequently observed in keloid scars and is considered a biomarker of keloids. Therefore, qRT-PCR was performed to study the effect of silibinin treatment on collagen type I alpha 1 chain (COL1A1) and collagen type III alpha 1 chain (COL3A1) gene expression in HDFs and KFs. In HDFs, TGF-β1 treatment induced the expression of COL1A1 and COL3A1 genes. TGF-β1-treated HDFs and KFs were treated with 100 or 200 μM of silibinin for 24 h and then examined for COL1A1 and COL3A1 gene expression at the mRNA level. As shown in Figure 5, silibinin significantly downregulated the transcription of both the COL1A1 and COL3A1 genes in a dose-dependent manner. 

In the TGF-β1-treated normal HDF, the relative expression of COL1A1 mRNA increased to 28.48 ± 5.413, while it decreased to 13.65 ± 5.693 and 6.268 ± 3.450 in HDFs treated with 100 and 200 μM of silibinin, respectively (Figure 5A). Similarly, the expression of COL3A1 mRNA in TGF-β1-treated normal HDF increased (1.152 ± 0.337) compared with that in the untreated control (1.0), while it decreased in HDF cells treated with 100 and 200 μM of silibinin (0.606 ± 0.178 and 0.268 ± 0.088, respectively; Figure 5B). In summary, treatment with 100 and 200 μM of silibinin resulted in a decrease of 52.1% and 78.0% in COL1A1 gene expression, respectively, as well as a decrease of 47.4% and 76.7% in COL3A1 gene expression, compared with TGF-β1-treated HDFs as a control (*p* < 0.001). 

In KFs, the relative level of COL1A1 mRNA transcript decreased to 0.672 ± 0.047 and 0.348 ± 0.043 after treatment with 100 and 200 μM of silibinin, respectively (Figure 5C). The relative level of COL3A1 mRNA transcript decreased to 0.608 ± 0.245 and 0.434 ± 0.282 after treatment with 100 and 200 μM of silibinin, respectively (Figure 5D). In summary, treatment with 100 and 200 μM of silibinin resulted in a decrease of 34.0% and 65.8% in COL1A1 gene expression, respectively, as well as a decrease of 39.5% and 56.7% in COL3A1 gene expression compared with the control in KFs (*p* < 0.001).

### 2.5. Silibinin Effectively Suppressed the mTOR Signaling Pathway in HDFs and KFs

To investigate the effect of silibinin on the mTOR signaling pathway, both KFs and HDFs were treated with 100 μM silibinin for 24 h. The key effectors in the mTOR pathway that are involved in protein synthesis, including p70S6K and 4E-BP1, were assessed by Western blotting. As shown in Figure 6A, TGF-β1 induced the phosphorylation of mTOR, p70S6K, S6, and 4E-BP1 in HDFs. KFs expressed abundant p-mTOR, p70S6K, S6, 4E-BP1, and their phosphorylated forms. The phosphorylated forms of mTOR, p70S6K, S6, and 4E-BP1 were notably reduced by silibinin treatment in TGF-β1-treated HDFs and KFs. 

Densitometric analysis showed that silibinin treatment resulted in a significant decrease in the p-mTOR/mTOR ratio in both HDFs and KFs compared with that in the control (55.4% and 21.0%, respectively, *p* < 0.01 and *p* < 0.05; Figure 6B). In HDFs, the decrease in phosphorylation after silibinin treatment was calculated using TGF-β1-treated HDFs as controls. In addition, silibinin decreased the phosphorylation of p70S6K by 36.5% and 26.8% in HDFs and KFs, respectively, as well as that of its downstream molecule S6 by 42.6% in HDFs and 21.2% in KFs compared with the control. 4E-BP1, another key downstream molecule of mTOR, showed decreased phosphorylation following silibinin treatment by 54.4% in HDFs and 12.3% in KFs compared with that in the control (Figure 6C–E). These results suggest that silibinin effectively suppresses the mTOR signaling pathway in HDFs and KFs.

## 3. Discussion

The molecular mechanisms underlying keloid scarring have been investigated for several decades. Several studies have demonstrated the upregulation of the mTOR signaling pathway in KFs and keloid tissues compared with normal controls [6,7,14]. mTOR is a serine/threonine protein kinase of the phosphoinositide 3-kinase (PI3K)-related kinase (PIKK) family that forms a subunit of two distinct protein complexes: mTOR complex 1 (mTORC1) and 2 (mTORC2) [15]. mTORC1 plays a central role in cell growth and metabolism by regulating the balance between the production of proteins, lipids, and nucleotides and the suppression of catabolic pathways, such as autophagy [5]. Meanwhile, mTORC2 controls cell proliferation and survival [5]. 

Studies have shown that mTOR is activated by the phosphorylation of Thr2446/Ser2448 in response to intracellular ATP levels and extracellular signals from nutrients and growth factors [16,17]. mTORC1 promotes protein synthesis by phosphorylating two key effectors: p70S6K and 4E-BP. Activated p70S6K plays a critical role in protein synthesis via phosphorylation of the 40S ribosomal protein S6 [5]. p70S6K phosphorylates S6 at multiple sites, including serine residues 235, 236, 240, and 244, to enhance the 5′-cap-dependent translation [18]. 

4E-BP inhibits translation by binding and sequestering eukaryotic translation initiation factor-4E (eIF4E) to prevent the assembly of the eIF4F complex. Cytoplasmic eIF4E promotes the initiation of 5′-cap-dependent translation, and nuclear eIF4E stimulates the nuclear-cytoplasmic transport of certain mRNAs [19]. mTORC1 phosphorylates 4E-BP at multiple sites, including threonine residues 37, 46, and 70, and serine residue 65, to trigger its dissociation from eIF4E, allowing mRNA translation [20,21] (Figure 7). 

Rapamycin, an mTOR inhibitor, binds to immunophilin FK506 binding protein-12 (FKBP12) to form the FKBP12–rapamycin complex. This complex specifically binds to mTOR and inhibits its kinase activity, thereby preventing the phosphorylation of p70S6K and 4E-BP1 [22]. Genome-wide microarray analysis has shown the impaired expression of collagen biosynthesis genes in scar-derived fibroblasts following rapamycin treatment [23]. Additionally, rapamycin suppresses the mTOR signaling pathway, and small interfering RNA downregulates the expression of ECM components, such as collagen in HDF [6,8,14]. These data suggest that pathological scarring can be modulated by inhibiting mTOR signaling pathways. 

Silibinin is a natural polyphenolic flavonoid isolated from the fruits and seeds of milk thistles [9]. Traditional milk thistle comprises approximately 65–80% silymarin and 20–35% fatty acids. Silymarin is a mixture of seven flavonolignans (silybin A, silybin B, silychristin, isosilychristin, silydianin, isosilybin A, and isosilybin B) and one flavonolol (taxifolin). Silibinin is a semi-purified silymarin comprising silybins A and B in a 1:1 ratio [9,24]. Silibinin inhibits the mTOR signaling pathway in various cancer cell lines [12]. Therefore, we investigated the effect of silibinin on collagen expression in both HDFs and KFs and its impact on the mTOR signaling pathway. We demonstrated that silibinin decreased the mRNA expression of types I and III collagen in TGF-β1-treated HDFs and KFs. Furthermore, the downregulation of key effector molecules of the mTOR signaling pathway responsible for protein synthesis was accompanied by a silibinin-induced decrease in collagen expression (Figure 8).

These results are consistent with those of previous studies confirming that silibinin reduces type I collagen expression and blocks the activation of Smad2/3-dependent signaling pathways in HDFs [13]. As the TGF-β/Smad signaling pathway has been considered a key pathway in fibrosis, several studies have been performed to elucidate the crosstalk between TGF-β/Smad and mTOR signaling. In human mesangial cells, TGF-β treatment induces mTORC1 activity in a TGF-β receptor-dependent manner 26. In addition, the overexpression of Smad3 increases mTORC1 activity, whereas the knockdown of Smad3 using short hairpin RNA decreases mTORC1 activity [25]. Similarly, in human lung fibroblasts, P70S6K and 4E-BP1 phosphorylation depends on Smad3 [26]. These results suggest that the TGF-β stimulation of mTORC1 requires Smad3 and that mTOR might play a critical role in fibrosis. The precise mechanism of the crosstalk between the two pathways and their effects on collagen synthesis remain to be fully explored.

Limitations of this study include its preliminary nature. Despite the decrease in collagen expression and the downregulation of the mTOR signaling pathway, the underlying mechanism of action remains unclear. Further comprehensive investigations are needed to fill these knowledge gaps, and it is essential to conduct in vivo studies to comprehensively evaluate the safety and efficacy of silibinin. Moreover, potential limitations and challenges exist for the clinical use of silibinin. Its chemical structure is highly hydrophobic and non-ionizable, which results in poor water solubility, measuring less than 50 µg/mL [9]. This characteristic significantly impacts the bioavailability of silibinin and poses challenges when developing oral formulations or topical agents. In addition, side effects, including flushing, mild diarrhea, pruritus, and dysgeusia, have been reported in studies that evaluated silibinin as a systemic agent [27]. These side effects must be carefully weighed against the potential therapeutic benefits of silibinin for its clinical use.

In conclusion, this in vitro study investigated the effect of silibinin on collagen synthesis through the modulation of key molecules of the mTOR signaling pathway, including p70S6K, S6, and 4E-BP1. The results indicate that silibinin significantly reduced the expression of collagen types I and III in HDFs and KFs, possibly via downregulation of the mTOR signaling pathway. These findings suggest that silibinin holds potential as a promising therapeutic candidate for keloid scar treatment and contributes to a broader understanding of its mechanism of action in treating keloid scars.

## 4. Materials and Methods

### 4.1. Collection of Keloid and Normal Tissue Samples

Freshly excised keloid scar tissues were obtained from patients without preoperative treatment. Tissues were collected from surgical excisions for keloid treatment. Informed consent was obtained from the Department of Plastic Surgery at Yonsei University College of Medicine, Severance Hospital in Seoul, Korea. Additionally, residual normal human skin specimens that were left behind during free TRAM breast reconstruction surgery were obtained from the Department of Plastic Surgery after approval from the Institutional Review Board (IRB) of Severance Hospital (IRB numbers: 4-2021-0262, 4-2022-1513).

### 4.2. Cell Lines, Cell Culture, and Isolation of Fibroblasts from Keloid Tissues

The HDF cell line (C0135C) was obtained from Gibco (Grand Island, NY, USA). The KF cell line (CRL-1762) was obtained from the ATCC (Manassas, VA, USA). As commercially available KF cells were derived from a 35-year-old African woman, we used two more cell lines (KF1 and KF2) isolated from a Korean patient with keloid to overcome ethnicity limitations. All cells were cultured in Dulbecco’s modified Eagle’s medium (DMEM; Lonza, Walkersville, MD, USA) with the supplement of 10% fetal bovine serum FBS (Gibco) and 1% penicillin–streptomycin (Gibco) in a humidified 5% CO_2_/95% air atmosphere at 37 °C. 

The excised keloid tissues were washed three times in phosphate-buffered saline to isolate KFs. The epidermal tissue and subcutaneous fat were removed, and the remaining dermal tissue was cut into small pieces (5 × 5 × 5 mm^3^) using a scalpel. The tissues were then incubated in DMEM supplemented with 10% FBS and 100 U/mL penicillin–streptomycin. The cell line was incubated at 37 °C in a humidified 5% CO_2_/95% air atmosphere, and the media was exchanged once every 3 days. Cell growth was observed under an optical microscope (BX43; Olympus, Tokyo, Japan). After 2–3 weeks, when released fibroblasts were observed from the keloid tissue, the tissue was removed, and the cells were sub-cultured. Passage 0 (P_0_) cells were plated for subsequent cell culture and experiments, and the third to fifth passage cells (P3–P5) were used in subsequent experiments.

### 4.3. Natural Phytochemical

Silibinin (S0417) was purchased from Sigma-Aldrich (St. Louis, MO, USA), dissolved in dimethyl sulfoxide (DMSO) to make 100 mM stock solution, and stored at −20 °C under light-proof conditions until use. In all the experiments described below, the final concentration of DMSO did not exceed 0.1% (*v*/*v*), which is not toxic to HDFs and KFs.

### 4.4. Cell Viability Assay

In vitro cell viability and proliferation assays were performed using the Cell Counting Kit-8 (CCK-8; Dojindo, Kumamoto, Japan) to evaluate the cytotoxicity of silibinin in HDFs and KFs. Initially, cells were seeded at a density of 1 × 10^4^ cells/well in 96-well plates overnight and then treated with varying concentrations of silibinin (0–200 µM) at 37 °C for 24 h. The concentration of silibinin was determined by referring to previous studies [28,29,30]. After treatment, 10 μL of CCK-8 solution was added to the culture medium and incubated for 1 h. Optical density values were measured at 450 nm using an enzyme-linked immunosorbent assay (ELISA) microplate reader (Versa-Max; Molecular Devices, California, CA, USA), and cell viability was presented as a percentage of the control.

### 4.5. Histological Examination and Immunohistochemistry (IHC) Analyses

Keloid and normal tissues obtained via excision and TRAM were fixed in 10% formalin and embedded in paraffin. Paraffin blocks were cut into 4 μm thick sections, deparaffinized, and dehydrated using an xylene and ethanol series. Histological examinations were performed on formalin-fixed, paraffin-embedded skin tissue sections. According to the manufacturer’s instructions, tissue samples were stained using a hematoxylin and eosin (H&E) staining kit (ab245880; Abcam, Cambridge, MA, USA). Immunohistochemistry (IHC) was performed on consecutively sectioned paraffin-embedded skin tissue samples. First, slides were processed and subjected to antigen retrieval. Primary antibodies for phosphorylated-mTOR (p-mTOR) (dilution 1:100, 44-1125G; Invitrogen, Waltham, MA, USA) were applied to the samples. A peroxidase/3,3 N-diaminobenzidine tetrahydrochloride detection kit (K5007; Dako, Carpinteria, CA, USA) was used. Finally, the cell nuclei were stained with hematoxylin (s3309; Dako), and the tissues were fixed using a mounting solution. Tissues were photographed at 400× magnification using an optical microscope (BX43; Olympus, Tokyo, Japan). Areas of p-mTOR signaling in the dermis were measured using ImageJ software (version: 1.54d, National Institutes of Health, Bethesda, MA, USA).

### 4.6. Quantitative Reverse Transcription–Polymerase Chain Reaction (qRT-PCR)

HDFs were treated with TGF-β1 (10 ng/mL; T7039; Sigma-Aldrich) for 24 h or treated with TGF-β1 and silibinin for 24 h. KFs were treated with silibinin for 24 h. Quantitative reverse transcription–polymerase chain reaction (qRT-PCR) was performed to study the effects of silibinin treatment on COL1A1 and COL3A1 mRNA expression in HDFs and KFs. Total RNA was extracted using an RNAiso Plus kit (Takara Bio, Kusatsu, Shiga Prefecture, Japan) according to the manufacturer’s protocol and quantified using a NanoDrop 2000 spectrophotometer (Thermo Fisher Scientific, Waltham, MA, USA). After cDNA synthesis using the RNA to cDNA EcoDry™ premix kit (Takara Sake, Berkeley, CA, USA), mRNA levels were assessed using qRT-PCR and SYBR Green Master Mix (4309155; Promega Corporation, Madison, WI, USA) on a QuantStudio 3 Real-Time PCR System (Applied Biosystems, Foster City, CA, USA). The cycling reaction conditions were as follows: 95 °C for 10 min, followed by 40 cycles of denaturation at 95 °C for 15 s, 60 °C for 20 s, and 72 °C for 30 s. The specific primer pairs used in this study are listed in Table 2. The mRNA levels were calculated using the 2^-ΔΔCt^ method, and relative mRNA expression levels were compared between control and silibinin treatments in KFs [31]. Glyceraldehyde 3-phosphate dehydrogenase (GAPDH) was used as the housekeeping gene to normalize gene expression levels.

### 4.7. Immunoblotting Assay

HDFs and KFs cultured with or without silibinin and TGF-β1 for 24 h were harvested, and cellular proteins were extracted using radioimmunoprecipitation assay buffer (Cell signaling technology, Danvers, MA, USA) supplemented with a protease inhibitor cocktail (PPI 1015; Quartett, Berlin, Germany). The protein concentrations were measured using a bicinchoninic acid protein assay kit (Thermo Fisher Scientific). Next, approximately 25 µg/µL protein was subjected to sodium dodecyl sulfate-polyacrylamide gel electrophoresis (SDS-PAGE) (8% SDS-PAGE for mTOR and p-mTOR, 10% SDS-PAGE for p70S6K, p-p70S6K, S6, and p-S6, and 12% SDS-PAGE for 4E-BP1, and p-4E-BP1). The resolved proteins were transferred to a 0.45 µm polyvinylidene fluoride membrane (EMD Millipore, Danvers, MA, USA). The membrane was then blocked with 5% skim milk and incubated overnight at 4 °C with various primary antibodies (Table 3). Subsequently, each membrane was washed with Tris-buffered saline with Tween^®^ (TBST; Sigma-Aldrich) and incubated with the corresponding secondary antibodies (Table 3) for 2 h at room temperature. Finally, the antigen–antibody complexes were visualized using ImageQuant LAS-4000 Mini (Fujifilm Life Science, Tokyo, Japan).

Immunoblotting results were analyzed using ImageJ software (version: 1.54d) to quantify the expression of target proteins and normalized using GAPDH as a protein-loading internal control. The level of protein phosphorylation is presented as the ratio of the phosphorylated form to the total form and was reported relative to the control, which was set at a value of 1.0.

### 4.8. Statistical Analysis

All experimental data are presented as the mean ± standard deviation (SD), and each experiment was conducted at least three times (*n* ≥ 3). Statistical analyses were performed using SPSS software (version 25.0; IBM Corp., Armonk, NY, USA). Statistical significance between two groups was calculated using Student’s t-test, and those between multiple groups were calculated using a one-way analysis of variance (ANOVA) followed by Tukey’s post hoc test. Differences with a *p*-value of <0.05 were considered significant.

## Figures and Tables

**Figure 1 ijms-24-14386-f001:**
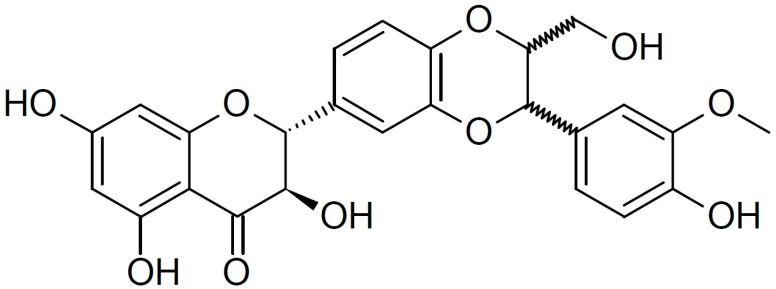
Chemical structure of silibinin. The chemical formula of silibinin is C_25_H_22_O_10,_ and the molecular weight is 482.441 g/mol.

**Figure 2 ijms-24-14386-f002:**
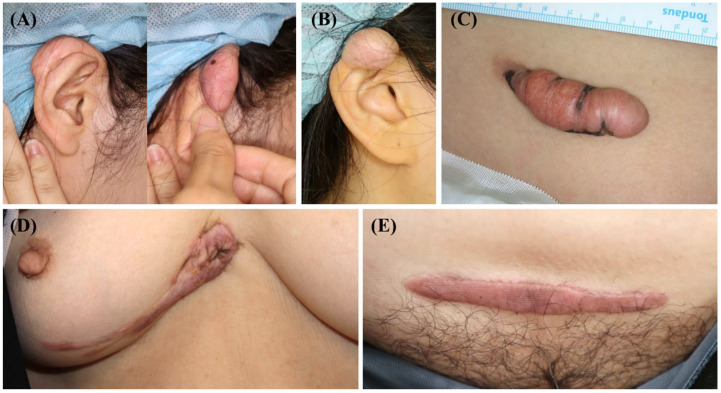
Clinical photographs of the keloid scars included in the present study. (**A**,**B**) Keloid scars on the right ear helix that occurred after piercing. (**C**) A keloid scar on the abdomen that occurred after an orchiopexy surgery. (**D**) A keloid scar on the anterior chest that occurred after a laceration. (**E**) A keloid scar on the abdomen that occurred after a total abdominal hysterectomy and bilateral salpingo-oophorectomy.

**Figure 3 ijms-24-14386-f003:**
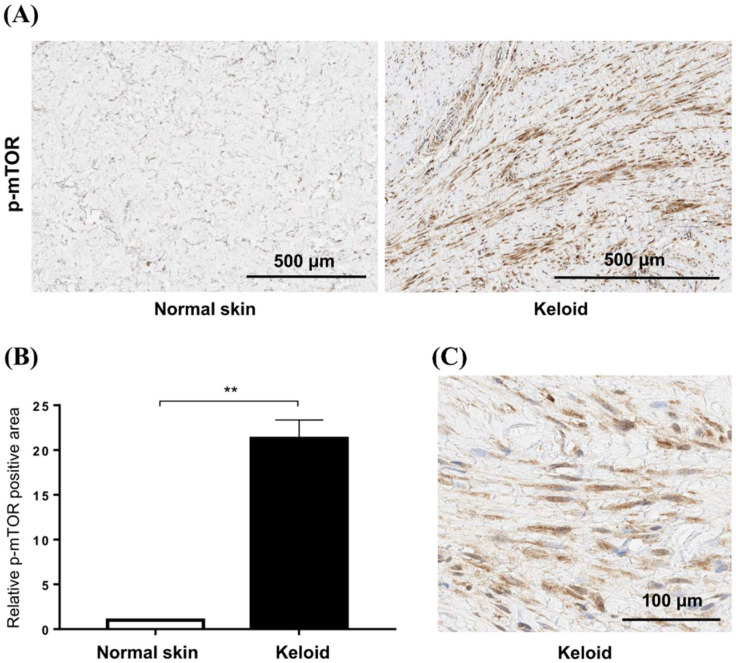
Expression of mTOR in keloid tissues. (**A**) Phosphorylation of mTOR is increased in active keloid scars compared to normal tissue specimens. The scale bar indicates 500 μm. (**B**) Semi-quantitative analysis showed that the p-mTOR signaling was increased by 21.3-fold in keloid tissue compared to normal tissue (** *p* < 0.01). (**C**) mTOR phosphorylation is increased in the cytoplasm of fibroblasts in keloid tissue.

**Figure 4 ijms-24-14386-f004:**
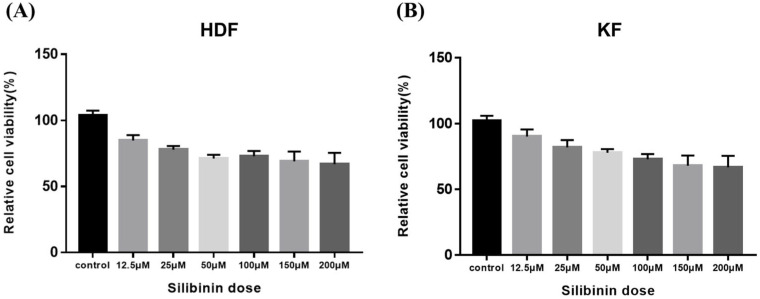
Effect of silibinin on cell viability of skin fibroblasts. (**A**) Human dermal fibroblast. (**B**) Keloid-derived fibroblast. Cells were seeded at a density of 1 × 10^4^ cells/well in 96-well plates overnight and then treated with different concentrations of silibinin (0–200 µM) at 37 °C for 24 h. Cell viability was analyzed using the CCK-8 assay kit. The viability of the cells was not markedly suppressed by treatment with silibinin at a concentration of 200 μM. HDF, human dermal fibroblast; KF, keloid-derived fibroblast; CCK-8, Cell Counting Kit-8.

**Figure 5 ijms-24-14386-f005:**
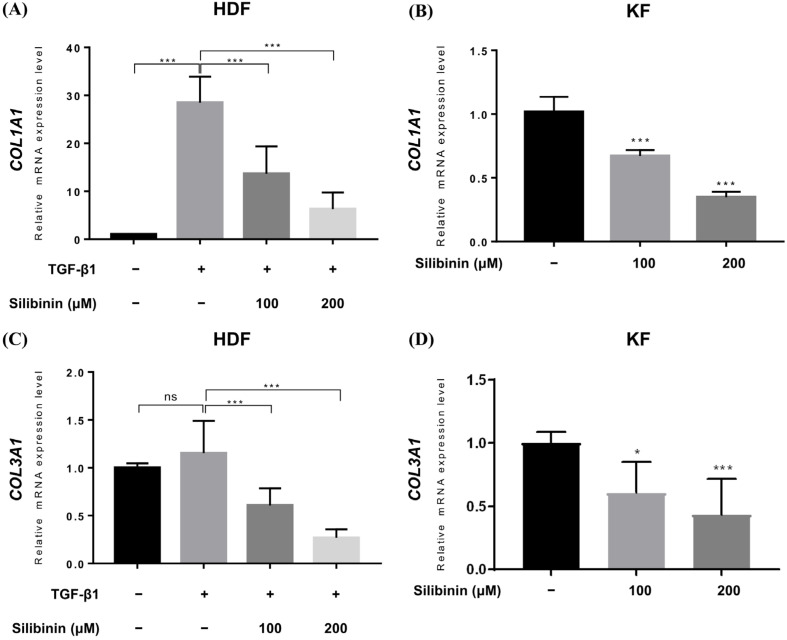
Effect of silibinin on collagen types I and III expression. Relative mRNA expression levels were compared between control and silibinin treatments. Silibinin treatment resulted in a dose-dependent reduction in COL1A1 (**A**,**B**) and COL3A1 (**C**,**D**) mRNA transcripts in TGF-β1-treated human dermal and keloid-derived fibroblasts. Bars represent mean ± SD, *n* = 3 − 6 (* *p* < 0.05, *** *p* < 0.001, ns, non-significant).

**Figure 6 ijms-24-14386-f006:**
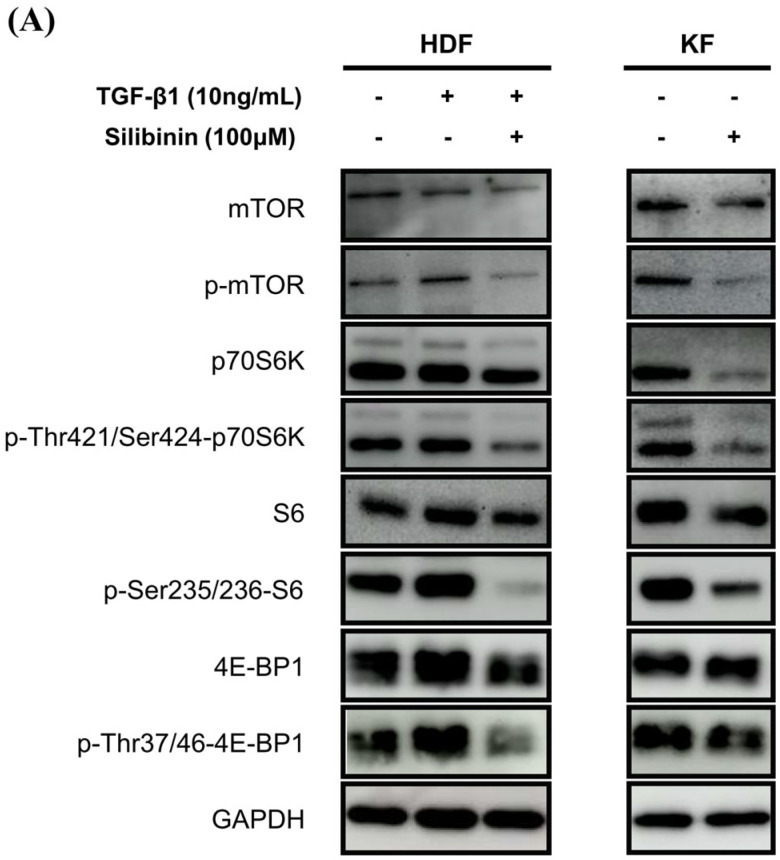
Effect of silibinin on the mTOR signaling pathway. (**A**) Silibinin effectively downregulated the mTOR pathway, including mTOR and its key components—p70S6K, S6, and 4E-BP1—in both HDFs and KFs. (**B**–**E**) Results of quantitative analysis. The data are presented as a ratio of the signal intensity of the phosphorylated form to the total form. *n* = 3 (* *p <* 0.05, ** *p* < 0.01, ns, non-significant).

**Figure 7 ijms-24-14386-f007:**
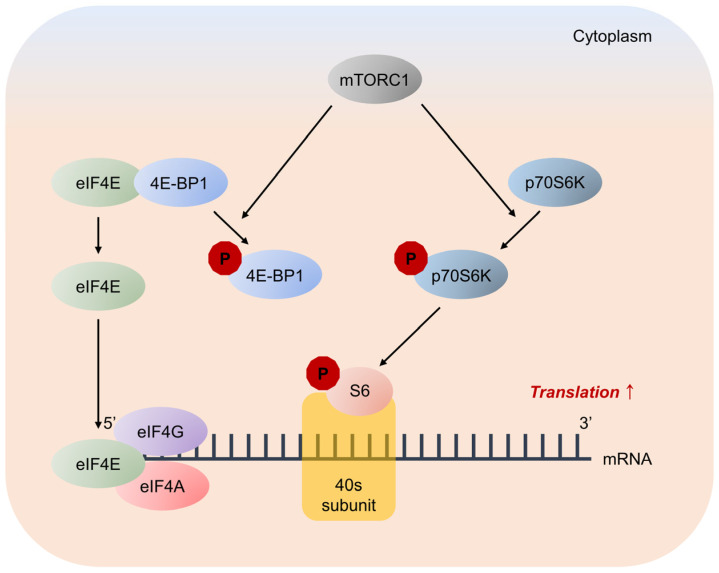
Major pathways downstream of mTORC1 signaling in mRNA translation. mTORC1 promotes protein synthesis by phosphorylation of two key effectors, eIF4E binding protein 1 (4E-BP1) and p70S6 Kinase (p70S6K). 4E-BP1 inhibits translation by binding and sequestering eukaryotic translation initiation factor-4E (eIF4E) to prevent assembly of the eIF4F complex. eIF4F complex mediates the recruitment of ribosomes to mRNA, which is the rate-limiting step for translation. mTORC1 phosphorylates 4E-BP1 at multiple sites to trigger its dissociation from eIF4E, allowing 5′-cap-dependent mRNA translation to occur. Unrelated to 4E-BP1, mTORC1 phosphorylates p70S6K, stimulating its subsequent phosphorylation. p70S6K phosphorylates and activates several substrates that promote mRNA translation initiation, including S6.

**Figure 8 ijms-24-14386-f008:**
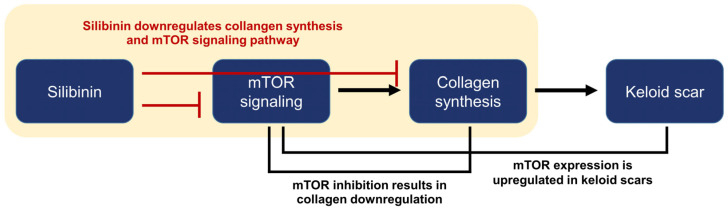
Summary and logic flowchart of the study. Silibinin treatment significantly reduced the expressions of collagen I and III in normal human dermal fibroblasts and keloid-derived fibroblasts, with suppression of the mTOR signaling pathway (highlighted in the yellow box). The relationship between mTOR, collagen synthesis, and keloid is described in the introduction and discussion sections of the manuscript.

**Table 1 ijms-24-14386-t001:** Clinical information of investigated tissue samples.

	Sex	Age (y)	Site	Duration of Scar (y)	Recurrence after Surgical Excision	Experiments Involving Tissue Sample
Participants with keloid scars						
1	F	25	Right ear	1	Yes	Cell isolation, IHC
2	F	29	Right ear	2	No	Cell isolation, IHC
3	M	11	Abdomen	2.5	No	IHC
4	F	76	Anterior chest	1	Yes	IHC
5	F	51	Abdomen	2	Yes	IHC
Donors of normal skin						
1	F	40	Abdomen	-		IHC
2	F	38	Abdomen	-		IHC
3	F	55	Abdomen	-		IHC

IHC, Immunohistochemistry.

**Table 2 ijms-24-14386-t002:** Sequence of primers used for qRT-PCR.

Target Gene	Primer Sequences (5′ → 3′)
COL1A1	Forward: 5′-TGTTCAGCTTTGTGGACCTCCG-3′ Reverse: 5′-CCGTTCTGTACGCAGGTGATTG-3′
COL3A1	Forward: 5′-GAAGATGTCCTTGATGTGC-3′ Reverse: 5′-AGCCTTGCGTGTTCGATAT-3′
GAPDH	Forward: 5′-CATGAGAAGTATGACAACAGCCT-3′ Reverse: 5′-AGTCCTTCCACGATACCAAAGT-3′

qRT-PCR, quantitative reverse transcription polymerase chain reaction; COL1A1, collagen type I alpha 1 chain; COL3A1, collagen type III alpha 1 chain; GAPDH, glyceraldehyde-3-phosphate dehydrogenase.

**Table 3 ijms-24-14386-t003:** Antibodies used in the present study.

Antibody	Host	Clonality	Isotype	Dilution	Product Code	Source
Primary antibodies						
Collagen I	Rabbit	polyclonal	IgG	1:1000	NB600-408	Novus Biologicals
Collagen III	Rabbit	polyclonal	IgG	1:1000	Ab7778	Abcam
mTOR	Rabbit	polyclonal	IgG	1:500	PA5-34663	Invitrogen
p-mTOR (Ser2448)	Rabbit	polyclonal	IgG	1:1000	44-1125G	Invitrogen
4E-BP1	Rabbit	monoclonal	IgG	1:1000	9644	Cell Signaling Technology
p-4E-BP1 (Thr37/46)	Rabbit	monoclonal	IgG	1:1000	2855	Cell Signaling Technology
p70S6K	Rabbit	monoclonal	IgG	1:1000	9202	Cell Signaling Technology
p-p70S6K (Thr421/Ser424)	Rabbit	monoclonal	IgG	1:1000	9204	Cell Signaling Technology
S6	Mouse	monoclonal	IgG1	1:1000	2317	Cell Signaling Technology
p-S6 (Ser235/236)	Rabbit	polyclonal	IgG	1:1000	2211	Cell Signaling Technology
p-S6 (Ser240/244)	Rabbit	monoclonal	IgG	1:1000	5364	Cell Signaling Technology
GAPDH	Rabbit	monoclonal	NA	1:1000	2118	Cell Signaling Technology
Secondary antibodies						
anti-rabbit IgG	Goat	NA	NA	1:2000	7074	Cell Signaling Technology
anti-mouse IgG	Goat	polyclonal	IgG	1:2000	SC-2005	Santa Cruz Biotechnology

Novus Biologicals, Centennial, CO, USA; Santa Cruz Biotechnology, Dallas, TX, USA; Abcam, Cambridge, MA, USA; Cell Signaling Technology, Danvers, MA, USA; Invitrogen, Waltham, MA, USA; mTOR, mammalian target of rapamycin; 4E-BP1, eukaryotic translation initiation factor-4E binding protein; p70S6K, p70KDa S6 kinase; Ser, serine; Thr, threonine; GAPDH, glyceraldehyde-3-phosphate dehydrogenase; IgG, immunoglobulin G; NA, not available.

## Data Availability

Data are available from the corresponding author upon reasonable request owing to privacy and ethical restrictions.

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
