# Peer review of "Silibinin Downregulates Types I and III Collagen Expression via Suppression of the mTOR Signaling Pathway"

_ijms, 2023, doi:10.3390/ijms241814386_

Round 1
Reviewer 1 Report
Dear Editors,
The following study submitted to IJMS demonstrates that Silibinin inhibits collagen synthesis in human dermal fibroblasts by targeting the mTOR signaling pathway.
However, there are several major concerns that arise.
Major issues
- There is a problem with the western blot data that needs to be addressed. The gel bands are very clear but the quantification is far from convincing. The authors present two sets of data (n=2) in each gel in the supplemental figures with two bands that are not labeled. When we compare the first set results with the second, data appear to be contradictory; for example, if we take p-mTOR, the protein increases in the first set (left) where cells are treated with TGF-beta1, whereas it rather decreases in the second set (right). The same issue arises for the remaining proteins. Additionally, GAPDH is not stable and authors should perform at least three experiments for each protein and not two.
- In the methods section, the authors state that primary fibroblasts were isolated from keloid tissues. However, HDF and KF cell lines were used throughout the manuscript. Unless I’ve missed it, but the authors should be more clear with the data presentation and include a flow chart to better present the methods.
- To strengthen the results, fibroblasts from normal tissue surrounding the keloid scars of the same patient should be isolated and cultured alongside KFs throughout the different experiments. Using two different cell lines might result in important inter-variabilities dampening the observed effects of TGF-β1 and Silibinin.
- Keloids and hypertrophic scars are both types of excessive scar tissue that can develop after a skin injury. However, there are some key differences between the two. The authors use the two terminologies interchangeably which is incorrect. A better clinical presentation of investigated tissues should be conducted by a dermatologist.
- Large parts of the manuscript seem to be generated by AI, as detected by ZeroGPT and GPTZero. I don’t know the stance of the journal toward such issues but this matter had to be highlighted.
- In Figure 3, p-mTOR staining should be better characterized to check whether the increased expression is localized in fibroblasts or inflammatory cells.
- What is the rationale behind the used Silibinin concentrations in vitro (any references)? This should be stated in the methods section.
- The authors satate that “Silibinin effectively downregulated collagen synthesis by targeting the mTOR signaling pathway”. This remains merely speculative unless experimentally demonstrated. Silibinin inhibited collagen synthesis and affected the mTOR pathway, however a direct link between mTOR and the fibrotic phenotype of the cells was not demonstrated. This could be performed by using mTOR activators w/o Silibinin or testing Silibinin in mTOR-overexpressing fibroblasts.
- In the statistical analysis, ANOVA but not t-tests should be done for multiple comparisons. Suitable posthoc tests should be also performed.
Minor issues
- The titles of paragraphs 4.5 and 4.7 in the methods section seem erroneous.
- The bibliography should be more up-to-date.
Reviewer 2 Report
ijms-2514917, Silibinin downregulates type â… , â…¢ collagen expression by targeting mTOR signaling pathway by Sooyeon Choi et al.
Comments:
Abstract:
- The authors should consider specifying the type of study this is (e.g., in-vitro, clinical trial, etc.) for the readers to quickly grasp the nature of the investigation.
Introduction:
- While the authors have mentioned that silibinin has been recognized by the WHO for its hepatoprotective and antifibrotic properties, it would be beneficial to add a line about the safety profile or any known side effects of silibinin to help readers assess the viability of it as a treatment option.
- The hypothesis statement at the end is good but consider using more active phrasing. For example: "We hypothesize that silibinin inhibits collagen formation through the mTOR signaling pathway, potentially making it a viable therapeutic agent for scars."
Materials and methods:
- What was the rationale behind the chosen silibinin concentration range (0-200 μM)? Some insight into how these particular concentrations were decided could be beneficial.
- The title for section 4.5 seems misplaced. Should it be "Assessment of Cell Viability and Proliferation" instead? This section relates to the assessment of silibinin's cytotoxicity rather than the safety of a device.
- For histological staining and IHC, the authors should also include the specific time points at which these were performed after treatment with silibinin.
- Did the authors perform any negative control for immunohistochemistry, such as replacing the primary antibody with an irrelevant antibody of the same isotype?
- In the qRT-PCR experiment, information about the primer specificity check should be provided. Were the primers tested for potential off-target amplification? Were melting curves analyzed to ensure single-product amplification?
Results:
- In the legends of the figures, it would be helpful to state the statistical test(s) used for the analyses for clarification and include more information about the sample size and measures of variance (standard deviation, standard error).
Discussion:
- It would be beneficial to compare and contrast your findings with specific studies that have reported similar or contrasting results.
- The authors have clearly outlined the mechanism of mTOR pathway and its link to keloid scarring. However, it would be beneficial to provide a more detailed discussion on how silibinin specifically interferes with this pathway, perhaps providing a figure or schematic for clarity.
- It would be valuable to discuss the potential limitations or challenges in using silibinin clinically. For instance, discuss its bioavailability, potential side effects, or the necessity for further in vivo studies.
- Consider addressing potential limitations of your study. Discussing limitations openly can provide valuable context for interpreting the results and shows the scientific rigor of your work. It can also be a good starting point for suggesting future research directions.
Conclusions:
- The conclusion succinctly summarizes the findings. However, it would strengthen your argument to reiterate the key findings from the study, the novel contributions your study has made to the field, and how these results might impact future research or treatment approaches.
Minor editing of English language required
Round 2
Reviewer 1 Report
Dear Editors,
The authors addressed all the previous concerns. Regarding the causality issue, although it is a constraint of this study, the authors made thorough explanations of it in the limitation section.
I don't have any further queries.
Thank you
Reviewer 2 Report
Dear Authors,
Thank you for addressing each of the comments and suggestions I provided in my initial review.
Minor editing of English language required